# 3D Tissue Reconstruction and Generation for Single-Cell Spatial Transcriptomics using Neural Radiance Fields

## Abstract

Single-cell spatial transcriptomics (scST) is a groundbreaking technique that allows for the exploration of gene expression patterns, cell-cell interactions, and tissue organization at the single-cell level. Traditional approaches in scST reconstruction mainly focus on assigning two-dimensional (2D) coordinates to individual cells within a pre-established region. This often requires a large amount of 2D slice data, such as ssDNAs images, which escalates both costs and the complexity involved in studying and reconstructing the tissue's three-dimensional (3D) organization. Here, we introduce a novel method for scST reconstruction, which is a Neural Radiance Fields (NeRF)-based 3D-aware generative model termed STscan, that aims to reconstruct a 3D scST scene using a minimal amount from 2D images (fewer than 10). Additionally, STscan can identify cell types and their expression levels within this 3D environment. To the best of our knowledge, STscan is the first NeRF-based method specifically designed for single-cell ST reconstruction, and it is the first end-to-end solution capable of directly reconstructing in vitro cell-cell environments from ssDNA images. This approach has the potential to significantly reduce both the complexity and cost associated with scST studies.

## 1 Introduction

The growing scientific consensus posits that cellular spatial positioning has a profound influence on gene transcriptional activity, ultimately affecting physiological processes. Situated within complex three-dimensional microenvironments, cells occupy specific spatial coordinates and execute specialized functions(Li et al., 2022; Palla et al., 2022). For example, in the human brain, neurons in the hippocampus are spatially organized in a way that allows them to effectively process and store memories. This unique arrangement ensures that incoming signals are relayed through a specific network of neurons, optimizing the brain's ability to encode, store, and retrieve information(Piwecka et al., 2023). Advances in Single-Cell Spatial Transcriptomics (scST) technologies have opened new avenues for cellular observation(Longo et al., 2021). These technologies, which enable the simultaneous measurement of gene expression profiles at single-cell or even subcellular resolutions while preserving spatial information, were recognized as Nature Methods' Method of the Year for 2020(Marx, 2021).

Within the scope of scST technologies, methods can be divided into two primary categories: In Situ Hybridization (ISH) methods and Spatial Barcoding techniques. The raw data output of scST for each respective methodology amalgamates both imaging data (including, Staining Images are often referred to as ssDNA, Sequecial images) and localized sequencing information Figure 1, thereby facilitating the identification of cell types and aiding in the understanding of their expression patterns to gain insights into biological processes(Kleino et al., 2022; Dries et al., 2021; Tian et al., 2023).

While there have been rapid advances in technologies for handling standard dimensional reduction, clustering, and differential expression tools in spatial transcriptomic data, methods that effectively leverage the most crucial feature of profiling—space itself—have lagged far behind(Burgess, 2019; Bressan et al., 2023). To enable more general spatial profiling, some researchers employ a particular method: they produce serial thin sections from a biological sample, process each section through

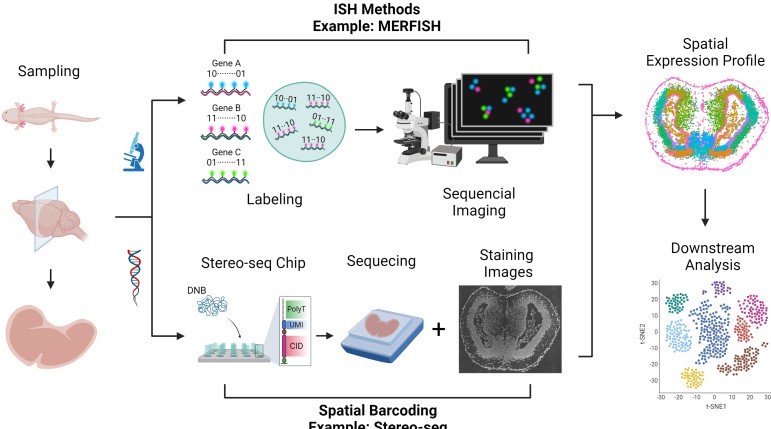

Figure 1: **The scope of scST methods.** After initial sample processing, both methods require the integration of sequencing with microscopic imaging for cellular data spacial extraction. The ISH approach often involves imaging sequencing results through fluorescent markers on the sequence. In contrast, the spatial barcoding technique presents a staining image (i.e., ssDNA) in the RNAscope procedures and decodes through 2D information. Ultimately, both methods aim to map cell types to their respective spatial expression levels, laying the foundation for downstream analysis.

2D imaging, and then use computational methods to realign the data and produce a 3D cube(Qiu et al., 2022; Xu et al., 2023). For example, Chen et al. (2023) used Stereo-Seq to reconstruct the monkey brain. They utilized 86 continuous slides in the monkey brain and recognized 143 macaque cortical regions. From this, they obtained a comprehensive atlas of 264 transcriptome-defined cortical cell types and mapped their spatial distribution across the entire cortex. They also discovered a relationship between the regional distribution of various cell types and the region's hierarchical level in the visual and somatosensory systems. However, this approach is very expensive and difficult to reproduce. Meanwhile, the loss of spatial information during the experimental process is an issue that stereo seq has yet to overcome.

With the goal of cost-effectively modeling the 3D structure of single-cell spatial transcriptomics (scST), our study employs neural radiance fields (NeRFs) (Mildenhall et al., 2021) as the 3D representation, termed the method **STscan**. NeRFs have recently made significant strides in view synthesis, they represent a 3D scene as a continuous radiation field parameterized by a neural network as inputs are coordinates and view directions. Despite showing promising results in common scenes and macroscopic objects, the problem of reconstructing microenvironments with NeRF remains largely unexplored. Therefore, we choose to apply them to the biomedical realms, specifically, we introduce a NeRF-based generator that renders and synthesizes novel views for single-cell spatial transcriptomics from a set of unposed ssDNA images.

Due to the data scarcity stemming from collecting ssDNA images through RNAscope procedures, we contend with a constrained quantity of training samples. To resolve this, we choose to jointly encode cell-type information and expression patterns along with appearance and geometry, which yield a joint distribution of ssDNA and corresponding semantics. This joint distribution affords us an enhanced understanding of the internal structure of scST and enables the modeling of the spatial distribution of cell type and expression, thereby assisting subsequent analyses of biological significance. By incorporating two discriminators with a differentiable data augmentation technique, we are able to synthesize high-resolution images while training our model solely on unposed 2D images. We systematically analyze our approach using raw data from the Stereo-Seq experiment(Wei et al., 2022). The experimental results demonstrate the efficacy of our model as a potent tool for scST image synthesis. Our model not only facilitates the reconstruction and generation of single-cell spatial transcriptomics (scST) from 2D images but also adds cell type and expression data simultaneously.

## 2  RELATED WORKS

**Application Insights in ISH and Barcoding Techniques**, in Situ Hybridization (ISH) represents a set of techniques specifically designed for tagging RNA molecules using fluorescent probes through complementary hybridization, which is subsequently visualized via fluorescence microscopy(Moses & Pachter, 2022). Among the available methodologies, three stand out based on their spatial resolution capabilities: MERFISH(Moffitt & Zhuang, 2016), seqFISH+(Eng et al., 2019), listed in descending order of resolution. Notably, MERFISH, a pioneering technique in this domain, was innovated by Professor Zhuang. Presently, it serves as a dominant technique in ISH applications, facilitating intricate studies on neurological structures(Zhang et al., 2021), oncological developments(Chen & Teichmann, 2021), and other related biomedical realms(Fang et al., 2022). In contrast, the realm of Spatial Barcoding techniques is characterized by the presence of unique barcoded DNA primers within each pixel, enabling the precise localization of a pixel in bidimensional representations. Noteworthily, this technique witnesses a broader application spectrum compared to ISH. Two of the paramount platforms in this field include 10x Visium(Galeano Niño et al., 2022) and BGI-Genomics's Stereo-Seq(Xia et al., 2022). However, given the limitation of 10x Visium in capturing single-cell granularity(Dong & Zhang, 2022), Stereo-Seq can capture in the subcellular emerging as a preferred choice, especially in contemporary research spanning genetics, oncology, and developmental biology(Koch, 2022). In light of these insights and the evolving research landscape, our current experiment harnesses data generated from the Stereo-Seq platform for reconstruction purposes.

**Cell Type and Expression Profiler**, in the realm of single-cell spatial transcriptomics, cell type identification, and cellular expression level analysis are two pivotal applications, laying the foundation for subsequent research and services(Cable et al., 2022). There are mainly two methods to estimate the cellular type composition: firstly, by assessing the enrichment level of cell type-specific markers in expressed genes, such as Leiden clustering(Traag et al., 2019); secondly, through deconvolution techniques aimed at precisely estimating the proportion of different cell types at each location, including SPOTlight(Elosua-Bayes et al., 2021) and DSTG(Song & Su, 2021). However, with the rapid advancement of spatial transcriptomics technology, the challenge of effectively integrating single-cell information with spatial transcriptome data has become increasingly prominent. Several groups are working on modeling spatial patterns of gene expression based on predefined processes(Bressan et al., 2023), like spatialDE(Svensson et al., 2018), whereas some methods mainly focus on spatial continuity. Incorporating continuous spatial expression information into spatial transcriptomic data also remains a challenge.

**NeRF**, short for Neural Radiance Field (Mildenhall et al., 2021), is a groundbreaking neural network architecture for 3D scene representation and reconstruction from 2D images. Since the inception of NeRF, many works have been proposed to enhance its quality and efficiency. For example, Yu et al. (2021); Xu et al. (2022a); Deng et al. (2022) aimed at diminishing the number of training views and enhancing generalization by utilizing image features or depth supervision. Barron et al. (2021; 2022) employ an integrated positional encoding of conical frustums to achieve anti-aliasing. Besides, some work combines NeRF with other tasks, such as Zhi et al. (2021); Fu et al. (2022) exploring incorporating semantic parsing with NeRF. Yariv et al. (2021); Oechsle et al. (2021); Wang et al. (2022); Li et al. (2023) have integrated NeRF with signal distance function and achieve both surface reconstruction and volume rendering using a single model. Furthermore, a series of studies have been conducted with the aim of augmenting the representation capabilities of NeRF through grid-based (Müller et al., 2022; Fridovich-Keil et al., 2022)and point-based (Xu et al., 2022b) architecture. Due to NeRF's generalizability, it has found applications across various domains, encompassing text-guided generation (Poole et al., 2022; Lin et al., 2023), human body modeling (Xu et al., 2021; Zhao et al., 2022), and even in the realm of medicine (Corona-Figueroa et al., 2022; Petkov, 2023). In this paper, we employ NeRF within the domain of microscopic medicine, specifically focusing on single-cell spatial transcriptomics. Our investigation is centered on the exploration of 3D reconstruction of scST with semantic information.

**Generative Modeling** endeavors to generate novel samples that manifest similar statistical properties to the training samples through learning the underlying distribution. During the early stage of deep learning, the variational autoencoder (VAE) (Kingma & Welling, 2013; Rezende et al., 2014; Kusner et al., 2017; Vahdat & Kautz, 2020) emerged as a popular generative model, which comprises an encoder network designed to map data into a latent space and a decoder network to reconstruct

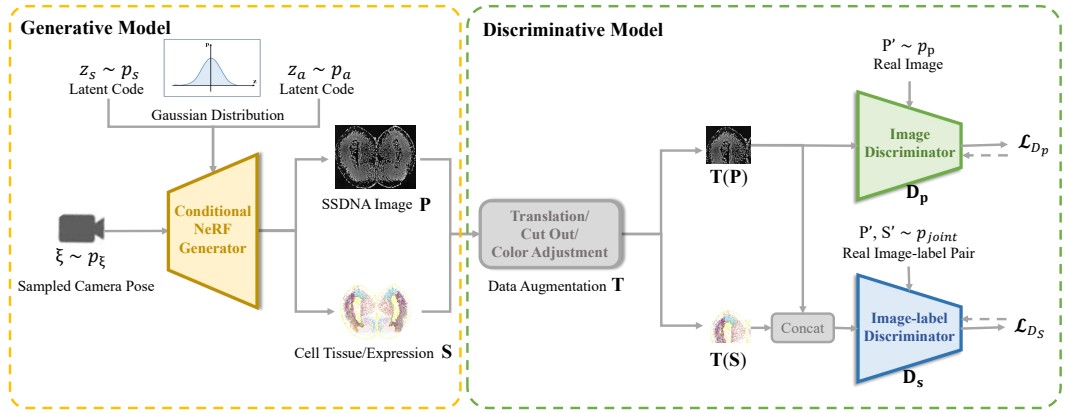

Figure 2: **The illustration of the overall network architecture.** Given a sampled camera pose $\xi$, the NeRF generator which is conditioned on two latent codes $z_s, z_a$ synthesizes an ssDNA image and corresponding cell type or expression segmentation. The synthesized results are first transformed by a differentiable transformation T for data augmentation and then fed into two discriminators $D_p$ and $D_s$, and the gradients of loss $\mathcal{L}_{D_p}$ and $\mathcal{L}_{D_s}$ will be backpropagated into the generator for optimization.

the data. Later, Generative Adversarial Networks (GAN) (Goodfellow et al., 2014) have become the predominant framework for generative modeling. It consists of a generator and a discriminator, which are trained in a competitive manner and have been applied to various applications including image synthesis (Brock et al., 2018; Qin et al., 2020), video generation (Tulyakov et al., 2018; Chu et al., 2020), style transfer (Azadi et al., 2018; Karras et al., 2019). Recently, diffusion models (Sohl-Dickstein et al., 2015; Rombach et al., 2022) have been proposed and achieved great success in image synthesis, they generate new samples by gradually denoising a normally distributed variable. However, there is a problem with all the above methods, they cannot generate 3D consistent scenes due to the lack of 3D modeling. Since NeRF has become one of the most popular 3D representations, many methods have been proposed to combine 2D generative models with NeRF and generate 3D assets. For example, Poole et al. (2022); Lin et al. (2023) incorporate NeRF with diffusion model and generate 3D objects using text prompt, while Graf (Schwarz et al., 2020) achieves 3d-aware image synthesis by integrating NeRF into GAN-based framework. In this paper, we adopt a GAN-styled framework based on Graf (Schwarz et al., 2020) and aim to generate new single-cell spatial transcriptomics under limited training data, to resolve which we introduce a joint distribution of scST and semantic labels and a data augmentation technique.

## 3 METHOD

Conventional 3D reconstruction methods for single-cell spatial transcriptomics (scST) are not only costly but also, due to their technical limitations, unable to recover lost information in the spatial domain. In our study, we seek to address the challenges of scST reconstruction and generation by employing computer vision algorithms, offering a cost-effective solution that can facilitate further research in the field of scST. In particular, we introduce a NeRF-based generative model, **STscan** that consists of three components: a conditional NeRF generator for synthesizing ssDNA images along with their corresponding semantic information, two discriminators for gradient backpropagation, and a data augmentation technique to address data scarcity. The overall network architecture is illustrated in Figure 2.

### 3.1 DATASETS GENERATION

We conduct **STscan** experiments on the brain of *Axolotl Telencephalon* using Stereo-Seq(Wei et al., 2022). During the utilization of this data, we preprocessed the cell types and their expression patterns. Current alignment methods cannot directly deal with images, so we employed a novel approach for **STscan**.

**Cell Type Images Generation**. For our study, processed Stereo-seq data was downloaded from ARTISTA, which encompassed the segmented cell bin matrix, cell coordinates, and annotation metadata. Key elements such as cell coordinates and annotations were extracted from this dataset. Following extraction, the data was using the ggplot2(Wickham, 2011), which is a plotting library. Finally, spatial maps for each section were crafted to depict the spatial distribution of cell types.

**Expression Pattern Images Generation**. Expression values for each gene were then stratified based on peak expression values observed in each section, resulting in categorization into six distinct levels: 0, Low, Below Average, Average, Above Average, and High. Following this stratification process, the processed data utilize the ggplot2(Wickham, 2011), and spatial maps for each section were generated, illustrating the spatial expression patterns of the selected genes.

**Affine Image Alignment**. To align ssDNA images and the images of cell types or expression patterns, we employed an affine transformation based on the optimization of key point alignments. A 2D affine transformation is represented as:

$$\begin{bmatrix} x' \\ y' \\ 1 \end{bmatrix} = \begin{bmatrix} a & b & c \\ d & e & f \\ 0 & 0 & 1 \end{bmatrix} \begin{bmatrix} x \\ y \\ 1 \end{bmatrix} \tag{1}$$

Our goal was to determine the optimal transformation parameters $a, b, c, d, e, f$ that minimize the Euclidean distance between transformed source points and target points. This optimization was achieved using the SGD gradient descent method(Qian et al., 2015), and the derived transformation was subsequently applied to the images.

## 3.2 GENERATIVE MODEL

We employ a NeRF as the representation for capturing the 3D structure of single-cell spatial transcriptomics inspired by Graf (Schwarz et al. (2020)). Besides, we leverage a discriminator $D_\phi$. to provide feedback to the generator and improve the realism of the synthesized ssDNA images.

**Conditional NeRF Generator**. NeRF is a neural network-based approach that has shown great success in 3D scene representation. It is designed to capture complex 3D scenes by learning a continuous radiance field parameterized by a neural network function, denoted as $G_\theta$, where $\theta$ represents the network's parameters. Specifically, given a 3D coordinate x and a viewing direction d:

$$G_\theta(x, d) = (c, \sigma), \tag{2}$$

where $\sigma$ is a volume density and c is the corresponding RGB color value c. Let r(t) = o + td denote a camera ray, the expected color C(r) of the ray with near and far bounds $t_n$ and $t_f$ is:

$$C(r) = \int_{t_n}^{t_f} T(t)\sigma(r(t))G_\theta(r(t), d)dt, \tag{3}$$

$$T(t) = \exp(-\int_{t_n}^{t_f} \sigma(r(s))ds). \tag{4}$$

With a sampled camera pose $\xi \sim p_\xi$, $G_\theta$ can generate a corresponding image patch P. Subsequently, the discriminator $D_\phi$ is employed to evaluate the synthesized patch P in comparison to an authentic patch P' extracted from the training dataset. During the training process, a 2D sampling pattern is applied to produce image patches at a resolution of K ×K for computational efficiency.

To achieve controllable generation, two latent codes are introduced, one of which is to model shape, denoted as $z_s \sim p_s$, and the other is to model appearance, denoted as $z_a \sim p_a$. Both $z_s$ and $z_a$ are sampled from Gaussian distributions. In particular, $z_s$ exerts control over shape by modulating the density $\sigma$, while $z_a$ operates on appearance. The formulation of the conditional NeRF is:

$$x, z_s \rightarrow \sigma, \tag{5}$$
$$x, d, z_s, z_a \rightarrow c, \tag{6}$$
$$G_\theta(x, d, z_s, z_a) = (c, \sigma) \tag{7}$$

**Semantic branch**. In contrast to usual macroscopic objects, single-cell spatial transcriptomics is a collection of microstructures, each falling into distinct cell types and different expressions. The

conventional NeRF formulation, primarily designed for capturing appearance and geometry, cannot model the inherent property distribution of cells. To address this limitation, **STscan** model a joint distribution of ssDNA and semantic labels by introducing a semantic branch that predicts cell type or expression labels and the formulation can be expressed as:

$$f_s(\mathrm{x}, z_s, z_a) = s, \tag{8}$$

$$\mathrm{S}(\mathrm{r}) = \int_{\mathrm{t_n}}^{\mathrm{t_f}} \mathrm{T(t)}\sigma(\mathrm{r(t)})\mathrm{f_s}(\mathrm{r(t)}, z_s, z_a)\mathrm{dt} \tag{9}$$

where s and $\mathrm{S(r)}$ are predicted semantic values, $\mathrm{f_s}$ is the branch for cell type or expression. The overall formulation for generative NeRF can be expressed as:

$$\mathrm{G}_\theta(\mathrm{x}, \mathrm{d}, z_s, z_a) = (\mathrm{c}, \mathrm{s}, \sigma) \tag{10}$$

The inclusion of semantic branches allows us to simultaneously acquire cell information during the synthesis of ssDNA images. This dual objective not only contributes to the reconstruction of ssDNA images but also improves the performance of generation by the acquisition of insights into cell type and expression distributions, which will be presented in the next section.

## 3.3  DISCRIMINATIVE MODEL

Utilizing the NeRF-based generator $\mathrm{G}(z_a, z_s) \to (\mathrm{P}, \mathrm{S})$, we achieve effective modeling of the joint distribution of ssDNA image patch P and the associated semantic attributes S, which encompass cell type or expression labels. These semantic labels inherently capture the intrinsic properties of ssDNA, therefore the joint distribution definitely enhances the generator's capacity to comprehend the 3D structure of ssDNA. In this section, we delve into a detailed exploration of strategies for leveraging the semantic attributes to provide feedback to the generator.

**Data augmentation**. As mentioned before, considering the difficulty of ssDNA data acquisition, **STscan** use a small set of training samples and focus on few-shot generation. However, the effectiveness of GAN is heavily dependent on the abundance of training data and the performance tends to deteriorate given a paucity of data. To resolve this, we introduce a data augmentation module following y DiffAugment (Zhao et al. (2020)). This module enhances data efficiency by applying a variety of differentiable augmentations to both authentic and synthetic samples. Specifically, we apply a random differentiable transformation T on both synthetic patch and real patch before feeding them into the discriminator, and T is a composition of three simple transformations including translation, cut out, and color adjustment. Building upon the data augmentation, we address the issue of inadequate training data while simultaneously mitigating concerns about discriminator overfitting.

**Discriminator**. We introduce two discriminators denoted as $\mathrm{D_P} : \mathrm{P} \to \mathbb{R}$ and $\mathrm{D_S} : \mathrm{concat}(\mathrm{P}, \mathrm{S}) \to \mathbb{R}$ respectively. Both discriminators are implemented with ResNet blocks. $\mathrm{D_P}$ plays the same role as the discriminator in most GAN models, it compares the predicted ssDNA patch P and the ground truth patch P', and the loss gradient is backpropagated to the generator thus facilitating the synthesis of more realistic images. $p_p$ denotes distribution over image patches of trainset, $\mathrm{f_D}$ means hinge loss and the loss function for $\mathrm{D_P}$ with data augmentation T is:

$$\mathcal{L}_{\mathrm{D_P}} = \mathop{\mathbb{E}}_{\mathrm{P'} \sim \mathrm{p_p}} [\mathrm{f_D}(-\mathrm{D_P}(\mathrm{T}(\mathrm{P'})))] + \mathop{\mathbb{E}}_{\mathrm{P}=\mathrm{G}(\mathrm{z_a}, \mathrm{z_s}, \xi), \mathrm{z_a} \sim \mathrm{p_a}, \mathrm{z_s} \sim \mathrm{p_s}, \xi \sim \mathrm{p_\xi}} [\mathrm{f_D}(\mathrm{D_P}(\mathrm{T}(\mathrm{P})))] \tag{11}$$

To leverage the semantic information, we also implement a discriminator $\mathrm{D_S}$ for identifying the semantic labels. $\mathrm{D_S}$ is dedicated to optimizing image-label pairs, whose input is the concatenation of P and S. This naturally ensures alignment between synthesized images and corresponding semantic labels, as the non-aligned image-label pairs can be easily classified into fake. The loss function for $\mathrm{D_S}$ is as follows:

$$\mathcal{L}_{\mathrm{D_S}} = \mathop{\mathbb{E}}_{P', S' \sim p_{joint}} [\mathrm{f_D}(-\mathrm{D_S}(\mathrm{concat}(\mathrm{T}(\mathrm{P'}), \mathrm{T}(\mathrm{S'}))))] \tag{12}$$

$$+ \mathop{\mathbb{E}}_{\mathrm{P}, \mathrm{S}=\mathrm{G}(\mathrm{z_a}, \mathrm{z_s}, \xi), \mathrm{z_a} \sim \mathrm{p_a}, \mathrm{z_s} \sim \mathrm{p_s}, \xi \sim \mathrm{p_\xi}} [\mathrm{f_D}(\mathrm{D_S}(\mathrm{concat}(\mathrm{T}(\mathrm{P}), \mathrm{T}(\mathrm{S}))))] \tag{13}$$

where $p_{joint}$ is the joint distribution of ssDNA and semantic labels. In addition, to generate better semantic results, we compute the L1 loss with respect to feature maps produced by semantic labels

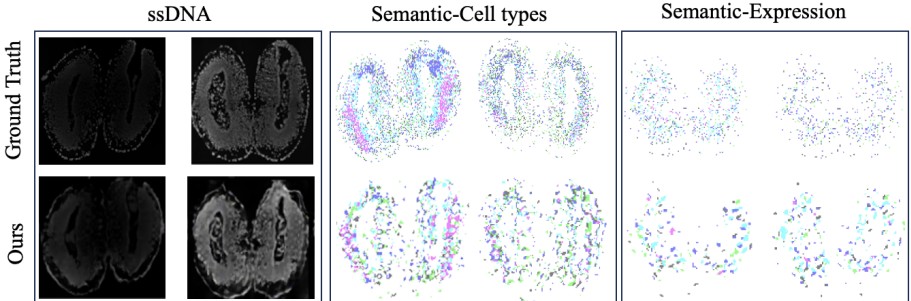

Figure 3: **Results of construct images.** The results of the 3D image generation process produced a set of images. These images were categorized into three distinct groups: ssDNA images, and two different sets of semantic images.

P and generated semantic P' in some hidden layers of $D_S$, and the objective of the generator is formulated as:

$$\mathcal{L}_G = \underset{P=G(z_a,z_s,\xi),z_a\sim p_a,z_s\sim p_s,\xi\sim p_\xi}{\mathbb{E}}[f_e(T(D_P(P)))] \tag{14}$$

$$+ \underset{P,S=G(z_a,z_s,\xi),z_a\sim p_a,z_s\sim p_s,\xi\sim p_\xi}{\mathbb{E}}[f_e(T(D_S(concat(P,S))))] \tag{15}$$

$$+ \sum_{i\in L} L_1(D_{S_i}(concat(P,S)) - D_{S_i}(concat(P',S'))) \tag{16}$$

Therefore, the total objective of the training procedure is defined as:

$$\mathcal{L} = \mathcal{L}_G + \mathcal{L}_{D_S} + \mathcal{L}_{D_P} \tag{17}$$

During the inference process, we randomly sample $z_s$, $z_a$ and camera pose $\xi$, and predict value and corresponding semantic information for all pixels in the image.

## 4 EXPERIMENT

### 4.1 RECONSTRUCTS THE BRAIN OF AXOLOTL TELENCEPHALON FROM SSDNA IMAGES

We embarked on our investigation by applying the **STscan** to axolotl telencephalon datasetsWei et al. (2022), considering both paired cell type information and developmental expression patterns. Our primary findings, as summarized in Table 1, the ssDNA reconstruction relied on two crucial metrics: the Peak Signal-to-Noise Ratio (PSNR) and the Structural Similarity Index Measure (SSIM), and the annotation including cell type and expression use mIoU metric. The generative loss function we employed showcased a notable perception-distortion trade-off in the renderings. Consistent alignment was observed between these renderings and both the cell type annotations and expression patterns, especially when viewed from continuous perspectives in comparison to the benchmark ground truth Figure3. For a more detailed evaluation, we noticed a consistency in the distance metrics associated with cell annotations and expressions. This consistency was further validated by comparing the cellular composition within pixels to their predecessor data. We specifically employed two metrics for this comparison: The Jaccard similarity coefficient is used to assess the similarity between cell types and their expression patterns in the original images compared to those in the reconstructed results, with a calculated result of 88.1%, and the Pearson Correlation Coefficient (PCC) is used to measure the similarity between cells in the reconstruction and their spatial distribution in the original images, with a calculated result of 90% Figure 6A. Both metrics manifested exemplary performance, suggesting that our model adeptly captures additional spatial information.

| Semantic type | ssDNA | | Semantic |
| --- | --- | --- | --- |
| | PSNR | SSIM | mIoU |
| Expression | 25.0294 | 0.6972 | 0.8456 |
| Cell type | 27.3329 | 0.6115 | 0.7792 |

Table 1: Quantitative results on reconstruction.

| Method | FID | KID |
| --- | --- | --- |
| w/ DiffAug | 82.4411 | 0.0544 |
| w/o DiffAug | 224.1103 | 0.1944 |
| w/o Semantic | 188.0024 | 0.1722 |
| w/Cell type | 81.2233 | 0.0609 |
| w/Expression | 79.1333 | 0.0588 |

Table 2: **Ablation study** on the effectiveness of semantic branch and data augmentation.

## 4.2 ABLATION STUDY

To validate the effectiveness of our proposed model, we conducted two sets of ablation studies: one focusing on the semantic branch and the other on data augmentation. The quantitative results are presented in Table 2.

As depicted in the table, the absence of the semantic branch results in a noticeable increase in both FID and KID scores for the generated samples (lower FID and KID values are indicative of better performance). This observation underscores the significance of incorporating a joint distribution encompassing both image and semantic information in enhancing the synthesis of ssDNA.

Additionally, we conducted an ablation study on data augmentation by simply removing the transformation T. The omission of data augmentation led to a substantial increase in both FID and KID metrics. This can be attributed to the limited size of our training dataset. In the absence of data augmentation, there is an elevated risk of discriminator overfitting, which adversely affects the quality of the synthesized samples. The introduction of diffAugmentation effectively mitigated this issue, resulting in improved generation results. Furthermore, Figure4 provides qualitative results to further support the above analysis.

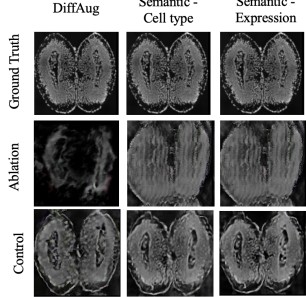

Figure 4: **The results of ablation experiments.** The control group consists of images generated by jointly encoding cell-type information and expression patterns, as well as applying data augmentation.

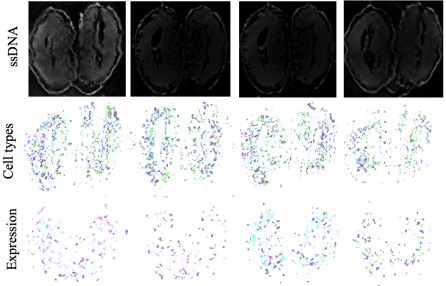

Figure 5: **Qualitative Results of Generative Reconstruction.** By leveraging the parameters $z_s$ and $z_a$ for inference control, STscan can generate a series of results, including both the ssDNA outcome and its corresponding cell type information or cellular expression.

| Method | FID | KID |
|---|---|---|
| w/Expression | 80.5758 | 0.0667 |
| w/ Cell type | 76.3329 | 0.0552 |

Table 3: Quantitative results on generation.

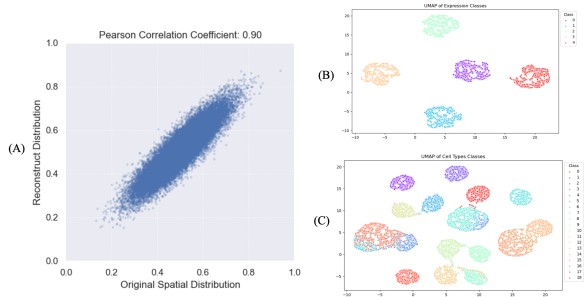

Figure 6: **The results of investigating biology meaning.** A represents PCC of cell distance between the reconstruction and 2D images. B and C summarize the UMAP results for cell type and expression. These results demonstrate a class type corresponding to the original classes.

### 4.3 EVALUATING GENERATIVE RECONSTRUCTION RESULTS

Our approach learns to disentangle continuous volumetric scenes which can be controlled via offering an inference control via parameters $z_s$ and $z_a$. Leveraging these adjustments, we engaged in synthesizing novel views within the scST datasets, as elucidated in Table II. The evaluation, grounded on the FID and KID metrics, and Figure5 provides qualitative results that support it.

Furthermore, to benchmark the consistency of our generative framework, especially in the biological domain, we implemented a cellular clustering approach on the synthesized scene outputs. By using UMAP for clustering the generated views, we found that our generated data maintained a high consistency with previous data in terms of both cell type count and expression categories Figure 6B,C. This attests to the robustness and precision of our proposed model in the realm of generative tasks.

## 5 CONCLUSION

In this study, we introduce a generative model based on Neural Radiance Fields (NeRF), **STscan** aimed at learning a continuous 3D representation for ssDNA images. To obtain additional biological information in a three-dimensional environment, such as cell types and expression profiles, we incorporated a semantic component into the model. Experimental validation revealed that the model achieves favorable results in both qualitative and quantitative reconstructions. Notably, we are the first team to introduce NeRF in reconstructing biological environments from a limited number of single-cell spatial transcriptomes, paving a new avenue in the field of biology and reducing the complexity and cost associated with scST research.

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

## A APPENDIX

You may include other additional sections here.

