# OpenReview forum: "3D Tissue Reconstruction and Generation for Single-Cell Spatial Transcriptomics using Neural Radiance Fields"
_ICLR.cc/2024/Conference — ICLR 2024 Conference Withdrawn Submission_

### Official Review · Reviewer_2SGX · 2023-10-28

**Soundness:** 2 fair
**Presentation:** 2 fair
**Contribution:** 2 fair
**Rating:** 3
**Confidence:** 5

**Summary:**

The paper presents a NeRF-based 3D-aware generative model for reconstructing a 3D spatial transcriptomics (scST) scene using a minimal number of 2D images while identifying cell types and their expression levels within this 3D environment. The proposed model consists of a conditional NeRF generator, two discriminators, and a data augmentation technique to address data scarcity. This approach has the potential to reduce both the complexity and cost associated with scST studies.

**Strengths:**

Single-Cell Spatial Transcriptomics (scST) technologies have garnered significant attention due to their extensive range of crucial applications. Utilizing NeRF-based generative models for this task is reasonable. Overall, the paper is well-written.

**Weaknesses:**

1. The novelty of the proposed approach is limited, as it primarily applies a NeRF-based generative model in scST without much adaptation.

2. Both the paper and the appendix lack a clear description of scST technology and a problem formulation that the approach aims to solve.

3. The experimental evaluation does not adequately support the application of the proposed method. It is tested only on the axolotl telencephalon dataset, raising questions about its generalizability. Moreover, the paper does not compare the proposed method with any baseline methods.

4. Section 3.1 omits crucial details, such as the number of images in the generated dataset.

5. The literature review could be improved by offering a more comprehensive overview of NeRF-based methods used in biological or medical image reconstruction tasks.

6. The caption of Figure 1 is unclear, as it does not specify which "both methods" it refers to.

Due to the ambiguity surrounding the problem being addressed and the absence of baseline comparisons, it is hard to determine the contribution of this work. The reported experimental results seem insufficient. These are the primary reasons I am inclined to recommend rejection.

**Questions:**

1. In Figure 3, there is a noticeable difference between the generated semantic images and the ground truth. Could the authors provide an explanation for these differences?

2. The paper does not incorporate any comparisons with baseline methods. Could the authors provide an explanation for this omission?

3. Would the authors be able to present a detailed problem formulation, which should ideally be placed at the beginning or before the method section?

---

### Official Review · Reviewer_TxGH · 2023-10-31

**Soundness:** 3 good
**Presentation:** 2 fair
**Contribution:** 2 fair
**Rating:** 5
**Confidence:** 4

**Summary:**

The authors developed STscan, a 3D-aware generative model based on Neural Radiance Fields (NeRF) for single-cell spatial transcriptomics (ST) reconstruction, by jointly encoding cell-type information and expression patterns facilitating image synthesis and reconstruction. The proposed method additionally has components for synthesizing ssDNA images, a semantic branch to predict cell types or expression labels, a GAN to evaluate the authenticity of synthesized patches and semantic attributes and data augmentation module to overcome data scarcity.

**Strengths:**

- The authors tackle the difficult, noteworthy challenge of 3D modeling of ST data with ssDNA images, which has not been sufficiently explored in the field.
- The choice of various components of the method are well-motivated. In particular, modeling the joint distribution of ssDNA images and semantic labels to effectively capture the intrinsic properties of ssDNA is well-thought through.

**Weaknesses:**

- The results of reconstruction in Fig 3 seems to show that the single cell structures are lost compared to the groundtruth data and the metrics point to the same issue. This brings into the question whether the method can effectively model single cell ST data.
- There is very limited rigorous biological evaluation of the proposed method. Their experiments focus on the Axolotl Telencephalon Stereo-Seq dataset, which is very limited in scope - it has smaller spots compared to 10x Visium, but still cannot be considered as single cell resolution compared to techniques like MERFISH or 10x Xenium.
- Although authors have employed several computer vision techniques in the paper applied to ST data, there is very limited technical novelty in the paper that makes it suitable for an ICLR submission (may be better suited for several other computer vision/ computational biology conferences).

**Questions:**

- The problem of reconstructing ST data from limited information in ssDNA images may be an ill-posed question. Imputation methods for sequencing data have been long explored and the field have settled on the understanding that they may be reliable for several rarely expressed genes.
- It is experimentally feasible to collect 3D single cell ST data, for example in:
https://www.biorxiv.org/content/10.1101/2023.07.21.550124v1
https://elifesciences.org/reviewed-preprints/90029
These datasets might be more relevant for evaluating the proposed method.
- How stable is the model training? With various components, very limited training data and the GAN, mode collapse might be an issue. The authors should provide the average reconstruction results across multiple runs for reproducibility.
- The codebase is not available - making it open-source would help with reproducibility efforts.

---

### Official Review · Reviewer_Vcft · 2023-11-02

**Soundness:** 2 fair
**Presentation:** 2 fair
**Contribution:** 2 fair
**Rating:** 3
**Confidence:** 3

**Summary:**

The paper proposes a 3D reconstruction of single-cell spatial transcriptomics with a NeRF model using less than 10 ssDNA images and their corresponding cell type and gene expression plots. The model generates can syntheise new views of the ssDNA and the cell types and gene expressions. They use Stereo-Seq data which is the ST technieque with highest resolution. The generative model proposed is trained with 2 discriminator functions. The discriminator for real/fake image and discriminator for real/fake image/expression and cell type paired data. The later, ensures that there is alignment between the generated images and corresponding semantic labels (i.e. gene expression maps and cell types). They evaluate the proposed method using similarity metrics such as PNSR, SSIM, and FID. In addition to mIOU on the cell types.

**Strengths:**

The 3D reconstruction of ST image and expression data using NeRF model is a novel idea and can have great impact.

**Weaknesses:**

I find that there are several details that are not well explained. Please find some observation followed by questions:

Observations:
1. In Figure 3, the prediction and ground truth look very different. In the semantic images one can visually see many mistakes. The results don't look very reliable. Also it doesn't look like individual cell coordinates can be inferred.
2. The function T is used twice with different meanings, as NeRF transmittance function and as augmentation function. This makes it is confusing to understand the method equations.
3. The authors mention that previous methods predict coordinates. However, they do not compare the coordinates dervied from their predictions to previous methods.

Questions:
1. A NeRF model is usually fit on a specific scene. Does that mean that for every scan a new model need to be generated? What is the training time per model?
2. The camera pos for 3D tissue reconstruction is different from 3D natural scenes. How is the camera pos sampled? What are the parameters that vary, is it only depth? Are the real views modified in any way to vary camera pos?
3. What is the total dimension of the ground truth data and the training and test data? How many 3D reconstructions were used in the evaluation and How many slices?
4. The authors mention that they need less than 10 slices. To be able to reconstruct the scene, were the training slices distributed in a specific way?
5. In Fig 2, the data augmentation function T seems to be only appliced on the fake data. Why is that?
6. What is the function fe in equations 14 and 15?
7. During training, is there a ground truth that the output is compared against or only the real/fake discriminators? If yes, which equation corresponds to that and what is the loss function used?
8. In the sentence "synthesizes novel views for single-cell spatial transcriptomics from a set of unposed ssDNA images.". what does 'unposed' mean? If there is no position info then how is camera pos is selected?


Minor:
  - The following sentence needs to be fixed "(including, Staining Images are often referred to as ssDNA, Sequecial images)"

**Questions:**

Please check the weaknesses section.